# DNA Barcoding and Species Classification of *Morchella*

**DOI:** 10.3390/genes13101806

**Published:** 2022-10-06

**Authors:** Wei Sa, Jinxia Qiao, Qiyuan Gao, Zhonghu Li, Qianhan Shang

**Affiliations:** 1State Key Laboratory of Plateau Ecology and Agriculture, Qinghai University, Xining 810086, China; 2Key Laboratory of Resource Biology and Biotechnology in Western China, Ministry of Education, College of Life Sciences, Northwest University, Xi’an 710069, China

**Keywords:** *Morchella*, DNA barcoding, molecular identification, phylogenetic relationship, species divergence

## Abstract

True morels (*Morchella*) are a well-known edible fungi, with economically and medicinally important values. However, molecular identification and species taxonomy of the genus *Morchella* have long been controversial, due to numerous intermediate morphologies among species. In this study, we determined the identification efficiency of DNA barcoding and species classification of 260 individuals from 45 *Morchella* species, on the basis of multiple nuclear DNA markers. DNA barcoding analysis showed that the individual DNA fragment has a lower resolution of species identification than that of combined multiple DNA markers. ITS showed the highest level of species discrimination among the individual genetic markers. Interestingly, the combined DNA markers significantly increased the resolution of species identification. A combination of four DNA genes (*EF1-α*, *RPB1*, *RPB2* and ITS) showed a higher species delimitation than that any combination of two or three markers. Phylogenetic analysis suggested that the species in genus *Morchella* could have been divided into two large genetic clades, the Elata Clade and Esculenta Clade lineages. The two lineages divided approximately 133.11 Mya [95% HPD interval: 82.77–197.95] in the early Cretaceous period. However, some phylogenetic species of *Morchella* showed inconsistent evolutionary relationships with the traditional morphological classifications, which may have resulted from incomplete lineage sorting and/or introgressive hybridization among species. These findings demonstrate that the interspecific gene introgression may have affected the species identification of true morels, and that the combined DNA markers significantly improve the resolution of species discrimination.

## 1. Introduction

True morels (*Morchella* spp.) belong to the Pezizales, Morchellaceae, and have significant economic and medicinal values [1]; they are recognized as one of the most prized edible mushrooms in the world [2]. According to the latest Index Fungorum, (http://www.indexfungorum.org/names/names.asp), accessed on 21 July 2020, a total of 352 records (including species, subspecies and varieties) of *Morchella* are listed. Most true morels are distributed in temperate regions of the northern hemisphere, and East Asia or China are considered to be the diversity center of *Morchella* species [3]. The genus *Morchella* was traditionally divided into three groups: black morels, yellow morels and semi-free capped morels [4,5,6]; the classification was based mainly on gross morphological features, such as the color and shape of the pileus, and the extension of the cap, etc. In addition, according to the fruiting bodies and stipes features of mature ascocarps, the blushing morels were divided into a fourth group that included three species, *M**orchella rufobrunnea* Guzmán and *F. Tapia*, *Morchella guatemalensis* Guzmán et al., and *M**orchella rigidoides* R. Heim [7]. However, due to morphological plasticity and intraspecific variability [4,5,6], the microscopic characteristics of *Morchella* at different developmental stages differ. The morphology and color of *Morchella* species are highly variable [8], which complicates their delimitation and characterization [9,10,11,12,13]. It is quite difficult to categorize and identify *Morchella* species on the basis of traditional morphological classification. Therefore, in order to improve species discrimination within *Morchella*, it is necessary to find a scientific and effective approach to precisely and accurately distinguish different species.

In recent years, with the rapid development of DNA sequencing technology and phylogenetic analysis, DNA barcoding has proven to be an effective method to establish species identification and molecular classification; it uses standardized, variable, easily amplified and short nucleotide sequence fragments in the biological genome [14,15]. The technology initially used mitochondrial gene cytochrome c oxidase I (COI) as the core for a global bio-identification system for animals [16,17]. In fungi, multiple nuclear gene markers, i.e., the large subunit of the nuclear ribosomal RNA (LSU), ribosomal small subunit (SSU), the nuclear ribosomal internal transcribed spacer (ITS), the translation elongation factor 1-*α* (*EF1-α*), RNA polymerase II largest subunit (*RPB1*) and the RNA polymerase II second largest subunit (*RPB2*), have been used in species classification [4,18]. The molecular phylogenetic analysis based on nearly complete SSU rDNA and partial LSU rDNA sequences confirmed the monophyly of the family Morchellaceae [18]. Meanwhile, on the basis of restriction enzyme analysis of the 28S ribosomal RNA gene, phylogenetic relationships of *Morchella* and its related genera, *Verpa* and *Disciotis* (Pezizales: Morchellaceae), were resolved [4].

Currently, some studies utilized the genealogical concordance phylogenetic species recognition (GCPSR) method, which involves a combination of multi-gene DNA sequences (ITS, LUS, *EF1-α*, *RPB1*, *RPB2*), to support the classification of *Morchella* into three major groups, including the Elata Clade, Esculenta clade and the Rufobrunnea clade [19,20,21,22,23]. According to traditional evolutionary analysis, the Rufobrunnea clade was considered to be the oldest branch, while the half-opened morel (*Morchella semilibera*) was classified as the Elata Clade. Additionally, because the phylogenetic species of this genus has not yet been clearly described, O’Donnell et al. applied the concept of molecular phylogenetic species to define the species within the genus *Morchella,* thus avoiding the mistakes of previous naming attempts, and abandoning the genus Latin binomials from before. “Group + number” was used to distinguish species, “*Mel*-n and *Mes*-n” represented different phylogenetic species of Black morel/Elata Clade and Yellow morel/Escalata Clade, respectively [19,21]. Du et al. (2012) studied the phylogenetic relationship and evolutionary history of *Morchella* based on four combined nuclear gene fragments (LSU, *EF1-α*, *RPB1*, and *RPB2*) [3]; it was suggested that East Asia (mainly China) was the diversity center of *Morchella*. Meanwhile, the GCPSR method was also applied to the classification and identification of *Morchella* in Turkey, in which 62 species were divided into 15 taxa [21,22].

Although molecular phylogenetic analysis provides strong support for the monophyly of *Morchella* [3,19], there is no unified, universal standard for the classification of *Morchella* species; moreover, many species cannot be clearly classified within *Morchella*. As a result of the morphological plasticity and introgressive hybridization of *Morchella*, the application with Latin binomials in these species is complicated and difficult [19]. Meanwhile, the evolutionary relationships and molecular dating of the genus *Morchella* are still controversial. Therefore, this project applied the concept of DNA barcoding to find an “identity” that was suitable for *Morchella* species. Multigene genetic markers were also used to construct a phylogenetic tree to accurately identify species limits, and analyze the historical evolution of *Morchella*.

## 2. Materials and Methods

### 2.1. Experimental Materials

Samples of *Morchella* were obtained from the Maco River Forest Farm and Xining of Qinghai province, China. Firstly, acquired materials were sterilized and dried. A bench was sterilized using UV for 30 min. After that, the medium was placed into a culture dish. Next, a stipe of each strain was cut into a piece that was 1–3 cm^2^, washed with distilled water, and then put into 75% alcohol for about 30 s. The sample was then rinsed with DD water 3 times after soaking (about 30 s). After fully absorbing water with clean filter paper, the sample was placed in a culture medium for cultivation. The medium was divided into four regions, each with a piece. Finally, the culture medium was placed in a constant temperature incubator at 25 °C. After the mycelium germinated and grew, it was transferred to a new medium for purification with a sterile environment.

### 2.2. DNA Extraction, PCR Amplication and Sequencing

Total genomic DNA was extracted from mycelia cultivated from pure cultures using the fungal DNA kit (Sangon Biotech, Shanghhai, China), according to the instructions of manufacturer. PCR amplification and sequencing of the four nuclear genes, ITS, *RPB1*, *RPB2* and *EF1-α*, were carried out (Table 1. The PCR reactions were conducted in a 20-microliter mixture system that contained 10.0 µL of 2 × Taq PCR mix (Runde, Xi’an, China), 1.0 µL of each primer (5 µmol/L), 7.0 µL of ddH_2_O and 1.0 µL of template DNA (30–50 ng). PCR amplification was conducted using a PTC-2000 thermal cycler (MJ Research, Deltona, FL, USA), and the PCR products were purified and sequenced by Sangon Biotech (Shanghai, China).

### 2.3. DNA Molecular Barcoding

A total of 260 sequences representing 45 species of *Morchella* were finally used in the evaluation of molecular barcoding. The *Morchella* species names, sources and accession numbers that corresponded to the sequences, are detailed in Appendix A. The geographical distributions of *Morchella* were visualized using ArcGIS v10.2 software (ESRI, Redlands, CA, USA) (Figure 1).

The single and combination sequences were aligned in MEGA 7.0 software [27], using its ClustalW application, and then corrected manually; all parameters were kept at their default settings. Insertions/deletions (inDels) and single nucleotide polymorphisms (SNPs) were calculated employing DnaSP v5.10.01 [28], after the alignment sequences were manually adjusted. MEGA 7.0 software [27] was used to group the sequences of four single genes (*RPB1*, *RPB2*, ITS and *EF1*-*α*) and combined genes. In order to select DNA barcodes that had excellent recognition ability, two different methods recommended by the CBOL Plant Working Group (2009) were applied to evaluate the recognition effect of DNA barcodes, PWG-distance and tree-building methods [29]. (1) PWG-distance method: the CBOL Plant Working Group (2009) recommended the PWG-distance method to calculate distances by pairwise alignment base substitutions; pairwise nucleotide genetic distances were based on the Kimura 2-parameter (K2P) model that was obtained from the MEGA7.0 software [27]. If the minimum interspecific distance of a species was larger than its maximum intraspecific p-distance, and there was no obvious overlap, then DNA barcode recognition was considered to be successful and excellent [29]. (2) Tree-building method: the tree building method, based on the P-distance model and Kimura 2-parameter model (K2P), was used to construct phylogenetic trees for each single marker and the combination markers with program MEGA 7.0 [24]; 1000 replicates were used to evaluate bootstrap support. If all the individuals of a species were a monophyletic group in the tree, it was considered a successful identification [30]. The ratio of the number of species that were successfully identified to the total number of species was considered to be the identification rate for DNA molecular barcoding [31].

### 2.4. Phylogenetic Relationship Analysis

The data sets of the 1578-base-pair, 264-taxon Morchellaceae (*EF1*-*α*, *RPB1*, *RPB2* and ITS) were analyzed, and the visual of phylogenetic tree was constructed via RAxML software [32] with 1000 bootstrap replicates, including combined fragments for four outgroups and 260 *Morchella* species. The GTR model of evolution was identified using jModelTest v2.1.4 [33] for the combined four-gene 264-taxon of the Morchellaceae. In addition, the phylogenetic tree based on maximum parsimony (MP) method was run in MEGA7.0 [27], and statistical supports were obtained by bootstrapping with 1000 replicates.

### 2.5. Diversification Time Estimates

We estimated the differentiation time of *Morchella* lineage with a Bayesian approach implemented in BEAST v1.8.0 [34], using the 1121-base-pair combined genetic data sets (*EF1*-*α*, *RPB1* and *RPB2*), and the published divergence time of Morchellaceae was used as the calibration point [19]. Eight sequences from the *Verpa* and *Disciotis* groups were used as outgroups. The models employed for each of the four partitions were the following: GTR for ITS and *tef1-α*, and TIM1ef for *rpb1* and *rpb2*. The GTR model was the best fit model of nucleotide substitution for the combined data set, according to prior model selection with jModelTest v2.1.4 [33]. In order to accommodate for rate heterogeneity across the branches of the tree [35], we used an uncorrelated relaxed clock model [36]. The Yule process was applied as the tree prior, while all other priors were set to the default. BEAST was run for 50 million generations, with 1000 sampling steps. Then, TRACER v1.5 was used to test effective sample sizes (ESS), which were considered to be obtained when ESS reached values >200 [37]. We burned in 10% of the trees with TreeAnnotator v1.8.0 [38]; the remaining trees were annotated to generate a maximum clade credibility (MCC) tree, including differentiation times and highest probability density (HPD) values to assess the statistical uncertainty of the divergence time estimates. FigTree v1.3.1 was used for editing and visualizing the MCC tree [39].

## 3. Results

### 3.1. Sampling and DNA Molecular Sequences

DNA molecular sequences of 260 individuals were finally used in the evaluation of DNA barcoding and species classification of *Morchella*, which included 13 newly sequenced samples from Qinghai province (China), and 247 available sequences downloaded from GenBank (Appendix A). We collected the molecular data sets of four universal nuclear gene markers, ITS, *RPB1*, *RPB2* and *EF1*-*α*. Geographic distributions of the sampled *Morchella* species were from Asia, North America, South America and Europe (Figure 1).

### 3.2. DNA Barcodes

The sequence data sets of four nuclear DNA fragments were obtained from 260 individuals of 45 species in *Morchella* (Appendix A). The sequence length of the single candidate DNA barcodes of *RPB1*, *RPB2*, ITS and *EF1*-*α* aligned, were 420 bp, 391 bp, 452 bp and 312 bp, respectively. ITS showed the highest single nucleotide polymorphism (SNPs) (125, 27.65%) and the highest mutation sites (78) among these individual gene regions. Compared with other DNA markers, *RPB2* was the most conservative, with the fewest mutation sites and insertion/deletion sites (0) (Table 2). We investigated the variability of the four DNA genetic markers for *Morchella* species, and all DNA regions showed higher genetic variabilities between than within species (Table 2). The nuclear ITS region showed the highest interspecific sequence divergence (13.47%), followed by *EF1*-*α* (7.70%), *RPB1* (6.73%) and *RPB2* (0.26%). In addition, the intraspecific variability was also higher for the ITS (0.62%) fragment. The *EF1*-*α* gene had the lowest intraspecific variation (0.40%) among all of the detected single-locus barcodes. The barcoding gap between interspecific and intraspecific distances was graphed on the basis of the K2P model for each individual gene as well as combinations of markers (Figure 2). The results demonstrated that the intraspecific and interspecific genetic distances of the combination of all four markers still overlapped, but the overlapping region was relatively short (Figure 2).

The two methods, PWG-distance and tree-building, were used to evaluate the differences in *Morchella* species discrimination power (Figure 3). In general, the species grouping into separate clusters in the phylogenetic tree with bootstrap support of >50% was considered to be a successful species identification. In the PWG-distance analysis, the *RPB2* (63.04%) gene fragment indicated the highest rate of resolution among single genes; *RPB1* and *EF1*-*α* showed the same discrimination, for 53.33%, while the ITS gene for 57.78%. The percentage of species distinguishment ranged from 48.89% (*RPB1* + ITS) to 77.78% (*RPB2* + ITS) for combinations of two markers, and *RPB1* + *RPB2* provided a successful species recognition of 68.89%. Furthermore, *EF1*-*α* + *RPB1* + *RPB2* showed the highest species recognition (88.89%) among the combinations of three nuclear markers, (Figure 3). Interestingly, the *EF1*-*α* + *RPB1* + *RPB2* (88.89% resolution) and *EF1*-*α* + *RPB1* + *RPB2* + ITS (88.89% resolution) combinations revealed the highest rates of successful species identification in the *Morchella* species.

For the evaluation of the species discrimination rate, two different parameter models (P-distance, K2P) were initially used in universal NJ-tree-building method. We determined the efficiency of four single DNA barcodes and eleven combined molecular barcodes (Figure 4). The current study suggested that the ITS gene had a relatively higher percentage of successful species discrimination based on the P-distance and K2P methods, which were 48.89% and 44.44%, respectively. Furthermore, *RPB1* + ITS, *EF1*-*α* + *RPB1* + ITS and *RPB1* + *RPB2* + ITS had the same species discrimination rate based on the two models in the combined sequences, which were 60.00%, 71.11% and 71.56%, respectively. Taking the K2P model as an example, the results indicated that the identification rate of individual DNA markers was between 35.56% and 44.44%, among which ITS provided the highest identification power (44.44%), followed by the other DNA barcodes *EF1*-*α* (42.22%), *RPB2* (40.00%) and *RPB1* (35.56%). In the analysis of discrimination rate of the two-locus combination barcoding markers, it was shown that recognition of the barcode *EF1*-*α* + ITS, *EF1*-*α* + *RPB1* and *RPB1* + *RPB2* were 57.78%, and that the resolution of the other fragments were 60.00% (*EF1*-*α* + *RPB2*, ITS + *RPB1* and ITS + *RPB2*). The identification efficiency of DNA barcoding ranged from 68.89% to 75.56% for combinations of three markers. The highest discrimination rate (75.56%) was for *RPB1* + *RPB2* + ITS, and the lowest was for *EF1*-*α* + *RPB2* + ITS (68.89%). Interestingly, the combination of four markers (*EF1-α*, *RPB1*, *RPB2* and ITS) showed the highest species delimitation (84.44%), more than any combination of two or three DNA markers. The species discrimination of a single DNA molecular marker was significantly improved by the combination of four gene markers, which was 8.88–15.55% higher than combinations of three-gene fragments.

### 3.3. Phylogenetic Relationship

We constructed the evolutionary relationship of *Morchella* species using the maximum parsimony and maximum likelihood methods, on the basis of the four nuclear protein-coding genes, *RPB1*, *RPB*2, *EF1*-*α* and ITS. MP and ML analyses of the full 260-taxon matrix were used primarily to identify major well-supported lineages, which included the Esculenta (yellow morels) and Elata (black morels) Clades, so that the species limits within each lineage could be further investigated using genealogical concordance phylogenetic species recognition (GCPSR) [23]. The results showed that the same topology was obtained between the two phylogenetic analysis methods (Figure 5 and Appendix A). In the phylogenetic tree, 260 samples of 45 *Morchella* species formed an independent monophyletic evolutionary clade with high bootstrap value, which was further divided into two large genetic lineages, including the Esculenta Clade (yellow morels) and the Elata Clade (black morels) (Figure 5). These two lineages formed the sister-group relationship with a high support rate.

Within the Elata Clade, three individuals of *M. capitata* and two individuals of *M. exuberans* clustered into a small evolutionary clade. *Mel*-13 and *Mel*-26 together grouped into an individual genetic lineage. The sampled individuals, 10QHXN, 14QHXN and 16QHXN in China, and *Mel*-6, clustered into a small group. *M. norvegiensis*, *Mel*-19 and other nine collected samples (China) were found to cluster together as one monophyletic lineage. Furthermore, we found that *Mel*-41 (*M.*
*owneri*), *Mel*-6, *Mel*-7, *Mel*-9 and *Mel*-10 of Elata Clade (black morels) clustered into a large genetic clade. Additionally, in the Esculenta Clade (yellow morels), the *Mes*-31 (*M. yangii*) and *Mes*-32 (*M. yishuica*) were sister groups. Intriguingly, we found that there were nested evolutionary relationships among species in the Elata Clade (black morels); two or more phylospecies, *Mel*-19 and *Mel*-33, grouped a clade; *Mel*-8, *Mel*-22 and *Mel*-28 were grouped together in the evolutionary tree.

### 3.4. Divergence Time of Morchella

The divergence time of *Morchella* species was estimated using relaxed molecular clock analysis in BEAST v1.8.0, employing combinations of three nuclear markers *RPB1* + *RPB2* + *EF1-α* (Figure 6). Sequences of eight non-Morchellaceae outgroup taxa were used to calibrate the Morchellaceae differentiation time, using the published divergence time estimation of Morchellaceae (ca. 243.36 million years ago, Mya) in the middle Triassic as an external calibration point [19].

The results indicate that the split time between *Morchella* and outgroups *Disciotis* and *Verpa* was approximately 243.61 Mya. The divergence of *Verpa* and *Disciotis* was dated to the early Cretaceous 140.1439 Mya [95% HPD interval: 86.6102–210.8646], and the Esculenta and Elata Clades differentiated approximately at 133.11 Mya [95% HPD interval: 82.77–197.95] in the early Cretaceous period. The divergence of Elata Clade species was much earlier than for the Esculenta Clade; evolutionary diversification of the Elata Clade was dated at 77.84 Mya [95% HPD interval: 44.26–116.97] in the late Cretaceous, followed by radiation of the Esculenta Clade in the middle of the Eocene 44.83 [95% HPD interval: 21.69–76.23] Mya. In addition, the analysis of two large clades showed that the Elata Clade of *Morchella* was further differentiated into different species; the differentiation time of one group was 49.30 Mya, and *Mel*-6, *Mel*-7, *Mel*-9, *Mel*-10 and *Mel*-41 were grouped into another clade that was dated at 53.70 Mya. Moreover, the Esculenta Clade was also divided into two genetic groups, with divergence times at 26.49 Mya and 31.06 Mya. Our results showed that approximately 86% (36/42) of lineage diversification within *Morchella* took place between the Miocene period and present, including the rapid diversification of most of the Esculenta Clade in late Miocene.

## 4. Discussion

### 4.1. Evaluation of Barcoding Efficiency of Morchella

The concept of DNA barcoding was first proposed by Hebert et al. (2003) [16]; it was a biological identification system that was based on intra- and interspecific variations, and applied relatively short standard DNA fragments with enough variation and easy amplification to achieve rapid and accurate discrimination and identification of species. In the current study, we applied the relatively universal candidate DNA barcodes (ITS, *RPB1*, *RPB2* and *EF1-α*) to identify the sampled individuals of *Morchella*. We successfully amplified and sequenced the four candidate barcodes. Generally, two prerequisites for evaluating ideal DNA barcoding were required: (1) intraspecies genetic differences are significantly less than interspecies variations, and a significant difference existed between them [33,40]; (2) the species of study are monophyletic to each other in the phylogenetic tree, that is, different individuals of the same species could be closely clustered together [16,32]. Thus, we applied two different methods (PWG-distance and tree-building methods), recommended by CBOL Plant Working Group, to evaluate the extent of success in species discrimination.

In our study, the nuclear ITS region showed the highest intraspecific and interspecific divergences (0.00062 and 0.1347, respectively), as well as the highest species identification (44.44%) among the single candidate DNA barcodes. These results were largely consistent with previous studies on *Morchella* species [25,41]. Thus, we suggested that the ITS marker should be incorporated into the core barcode marker for *Morchella* species. Meanwhile, the discrimination rate of any combination of markers was higher than that of single markers, and had higher species differentiation and interspecific identification in *Morchella* species. In particular, the resolution rates of a four-gene combination between tree-building and PWG-distance methods were as high as 84.44% and 88.88%, respectively. In contrast with four single sequences, the multi-locus combination had more effective information sites and more variation sites in *Morchella*, which clearly supported the monophyly of species. Therefore, combinations of DNA barcodes can greatly improve species discrimination; moreover, combinations have the potential and advantages for species identification. However, we found that some *Morchella* species could not successfully be identified, even when employing the combined nuclear DNA markers. For example, different species *Mel*-22, *Mel*-28 and *Mel*-8 formed a monophyletic group; *Mel*-19, *Mel*-33 and nine samples from Qinghai province clustered into a genetic clade. We inferred that the complex life cycle, high morphological plasticity and introgressive hybridization among species may have blurred species boundaries, which further led to difficulties with species identification and confusion with species relationships [19]. In recent years, some phylogenetic analyses based on mating genes were performed in different ascomycetes, and obtained a high level of species identification [42,43,44,45]. Therefore, we could integrate the sequences of different genetic backgrounds in order to improve species recognition rate in the future.

DNA barcoding technology is a powerful supplement to traditional morphological classification, and is an approach that has accelerated the pace of species classification and identification. Once the species that is analyzed by combining molecular and morphological characteristics is identified, it will strongly support the “identification card” of species, thus building a reliable identification system [46]. Our study on *Morchella* species provided a case study for barcoding these macrofungi in the future.

### 4.2. Phylogenetic Relationships of Morchella

Multi-locus phylogenetic analysis can clearly clarify the genetic relationships among *Morchella* species, and its greatest advantage is based on the concatenated data sets of different gene fragments that make the available amount of phylogenetic information greater than that of a single fragment. Moreover, the simulation analysis of a combined data matrix showed that even if the topological structure of a phylogenetic tree of a single gene locus was inconsistent, the correct phylogenetic analysis can be obtained [47]. Multiple genes are therefore important to aid in species recognition, and they are often used, instead of morphology, to identify evolutionary relationships of species [48]. As far as *Morchella* is concerned, even though it is morphologically variable to some extent, molecular studies are helpful for the preliminary identification of the taxa, and to understand evolutionary relationships [49]. In the current study, phylogenetic analysis showed that the sampled *Morchella* species formed an independent evolutionary monophyletic clade, which was further divided into two large genetic lineages, including the Esculenta Clade (yellow morels) and the Elata Clade (black morels) (Figure 5). These results were largely consistent with previous studies that were based on a few DNA molecular markers [3,19]. We found that the genus *Verpa* was closely related to *Morchella* in the phylogenetic tree. Some related studies found that *Mel*-6, *Mel*-7, *Mel*-9 and *Mel*-10 usually occurred in burned areas [50], while *Mel*-41 was mainly distributed in low-elevation ecological habitats in northern China [51]. However, the phylogenetic results found that *Mel*-41 (*M. owneri*) and *Mel*-6, *Mel*-7, *Mel*-9 and *Mel*-10, clustered into a genetic group, leading to speculation that *Mel*-41 may have existed in the burnt area; further research and investigation are necessary. Moreover, it was found that *Mel*-13 and *Mel*-26 were closely related with the unique Latin binomials *M. deliciosa* [51,52]. Additionally, phylogenetic analysis showed that *Mel*-19 and *M. norvegiensis* formed a genetic clade; actually, some studies found that *M. norvegiensis* was a more ancient name of *Mel*-19 [53]. In the Esculenta Clade (yellow morels), *Mes*-31 (*M. yangii*) and *Mes*-32 (*M. yishuica*) were sister groups, which were similar in morphology, and consistent with previous results [51].

Additionally, we found that the phylogenetic status of some *Morchella* species has been shifting, resulting in confusion of species classification. For instance, *Mel*-19 was nested with *Mel*-33, and *Mel*-8, *Mel*-22 and *Mel*-28 in the Elata Clade were mixed and cannot be distinguished, with species boundaries between them being ambiguous. From the perspective of the geographical distribution of *Morchella* species, both *Mel*-19 and *Mel*-33 were distributed in Xinjiang, China (see Appendix A), which may have been caused by gene introgression. Recent research reports have shown that two sympatric morel species in China have genetic characteristics of high inbreeding, prevalent clonality, limited local recombination and potential hybridization or horizontal gene transfer [54]. Interestingly, we found that the heterogeneous species also appeared to be confounded. Perhaps incomplete lineage sorting (ILS) was an important reason for this phenomenon, which, due to the extremely short species differentiation time, the polymorphisms of ancestral genes became randomly fixed in the differentiated species. Therefore, we speculated that hybridization and/or gene introgression and/or ILS among species may have been the cause of non-monophyletic groups within *Morchella*, blurring the boundaries between species. Whether there is gene flow or ILS still requires follow-up research and analysis to determine this. Moreover, genus *Morchella* belongs to heterothallic fungi, namely obligately outcrossing fungi that are self-sterile and usually require the participation of the opposite mating type partner to reproduce; only in fourteen black morel species has this been found [47,52,55]. The weak research progress in the reproductive modes and life history of *Morchella* species was attributed, on the one hand, to previous taxonomy confusion; on the other hand, the absence of molecular information and the absence of genomic sequence of *Morchella* resulted in classification confusion within genus *Morchella*.

True morel is a precious and rare medicinal material, whose complicated life history, evolutionary history and habitat have led to overlapping morphological characteristics and the formation of transitional species, resulting in blurred interspecies relationships and many closely related species. Therefore, clarifying the relationships between species can not only elucidate morel biology, but also provide a theoretical basis for the introduction of *Morchella* species, the scientific formulation of breeding strategies, and further development, utilization and protection of germplasm resources.

### 4.3. Divergence Time of Morchella

Our results supported that two major clades of *Morchella* and outgroups of non-Morchellaeae originated in the middle Triassic 243.61 Mya, and that there was an early- to mid-Cretaceous origin of the Esculenta and Elata Clades 133.11 Mya [95% HPD:82.73–197.95]. These age estimates are slightly different from those of previous studies [3,19]. Due to the occurrence of transgression in central North America in the middle Cretaceous, and the uplift of the Rockies in the late Cretaceous [56], the isolated environment and other geographical factors may have hindered genetic exchange and migration, which further caused the formation of two independent biological groups, the Elata Clade and the Esculenta Clade. In addition, the morel species of the Elata Clade evolved earlier than those of the Esculenta clade, and species of the Elata Clade gradually evolved in the late Cretaceous, while the Esculenta Clade expanded its species diversity gradually during the Eocene. Some studies found that most *Morchella* species spread to Asia through the Bering land bridge, while a few species spread to Europe through the North Atlantic land bridge [3,19]. The study results showed that approximately 86% (36/42) of *Morchella* lineage diversification occurred in the Miocene era, which may possibly be related to the cold climate in the middle and high latitudes of the northern hemisphere, as well as the uplift of the Qinghai–Tibetan Plateau [54]. Furthermore, fragmentation of ancestral ranges due to late Miocene aridification is hypothesized to have contributed to the relatively recent and rapid allopatric speciation of *Morchella*.

The climate of the Quaternary period and associated environmental changes also caused great changes within species that made diversity and geographical distribution relatively complicated [53,57]. This may have caused larger-scale migration of organisms, even survival disaster [58,59]. The harsh climate of the ice age led to concentrations of organisms in the refuge, and after the ice age, the organisms reappeared after spreading from the refuge to the warm climate. Thus, geographical isolation caused by refuges not only promotes species differentiation, but also serves as a conservation area for species diversity [60,61,62]. Based on some studies, many scholars have found and proved that North America and Europe were Quaternary glacier refugia, and important shelters for plant and animals [60,63]. For *Morchella* species, North America and Europe provide some protection as a refuge, mainly distributed in the Mediterranean coast of Europe, and in the west and east coasts of North America [50]. As far as China is concerned, the Quaternary glaciation did not cause serious damage. However, the change in microclimate, coupled with the special environmental heterogeneity and complex topography in China, may have provided beneficial refuge for many species, resulting in significant species differentiation in this period [53,62]; thus, it is possible to interpret East Asia or China as a center of species diversity and modern distribution of *Morchella*.

Currently, the reliable method to estimate and calculate the time of species differentiation is on the basis of the attenuation of fossil DNA and DNA comparison of similar species today. Fungi have no fossils, so it is difficult to accurately estimate the age of their differentiation due to the loss of their ancient reference materials. Therefore, it is only possible to retrospectively trace their origins and the times of differentiation of various species through the rate of genetic mutation, the aid of data models, and geological events to infer the results. The estimation of the divergence time of *Morchella* species in this study is based on the secondary calibration of the evolutionary time of the genus in previous studies. It is necessary to obtain a more accurate evolutionary history of the genus in order to find reliable fossils to use as references.

## Figures and Tables

**Figure 1 genes-13-01806-f001:**
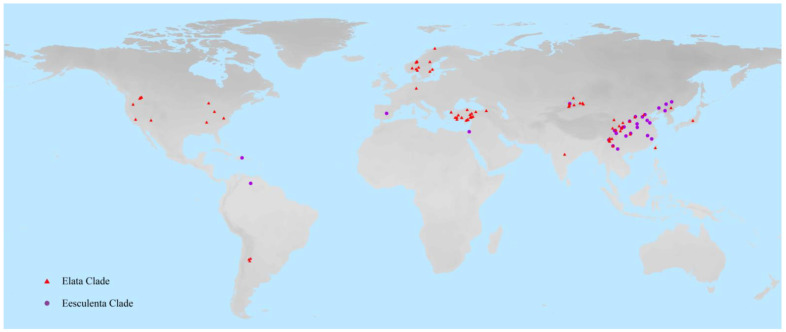
The sampled geographic locations of *Morchella* species. The red triangle indicates the geographical distribution of the Elata Clade (black morels), and the purple circle represents the Esculenta Clade (yellow morels).

**Figure 2 genes-13-01806-f002:**
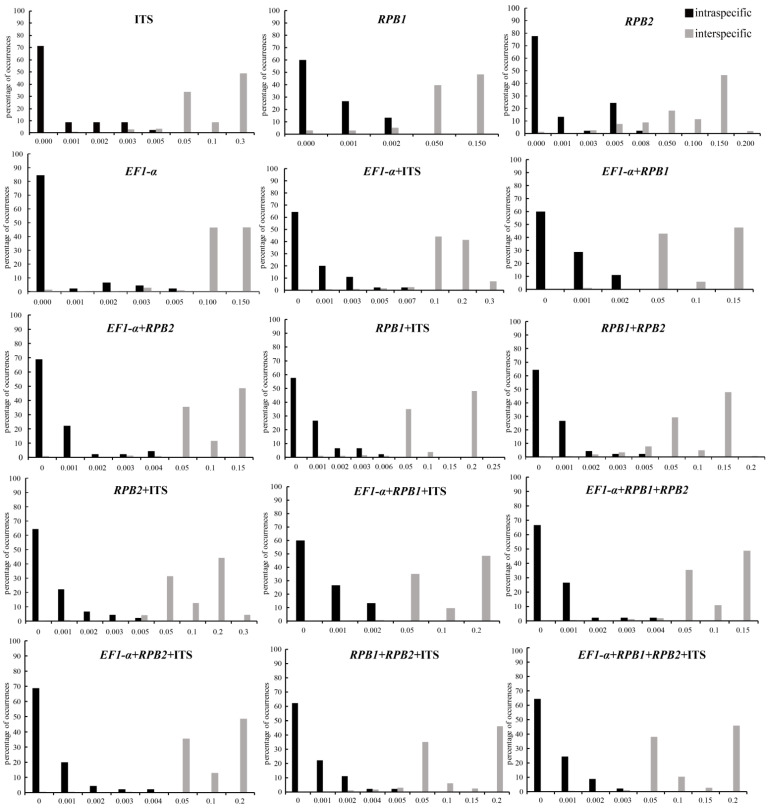
Histograms of the frequencies (*y*-axes) of pairwise intraspecific (black bars) and interspecific (gray bars) divergences based on the K2P distance (*x*-axes) for four single (ITS, *RPB1*, *RPB2* and *EF1-α*) and eleven candidate combination DNA markers.

**Figure 3 genes-13-01806-f003:**
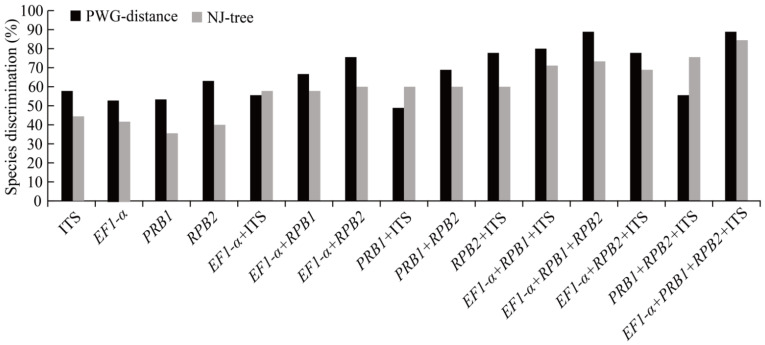
Statistics of species discrimination rates based on PWG-distance and NJ-tree methods. Histograms of the frequencies (*y*-axes) of the species discrimination rate based on PWG-distance and tree-building; *x*-axes represent 4 single- and 11 multi-locus barcodes in *Morchella*. The black bars express the resolution rate based on the PWG-distance approach, and the gray bars show the recognition power according to tree-building method.

**Figure 4 genes-13-01806-f004:**
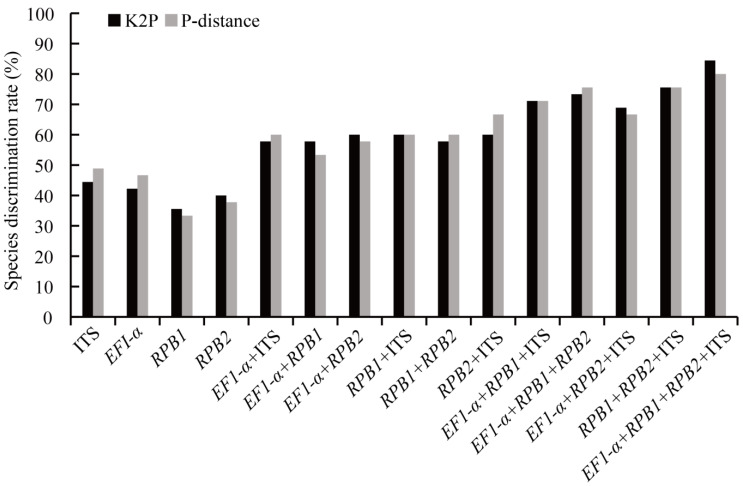
Species discrimination rate based on K2P and P-distance models in the NJ-tree approach. Sequences included all tested 4 single and 11 combined DNA fragments in *Morchella*. Histograms of the frequencies (*y*-axes) of the species discrimination rate based on K2P (black bars) and P-distance (gray bars); *x*-axes represent single- and multi-locus barcodes in *Morchella*.

**Figure 5 genes-13-01806-f005:**
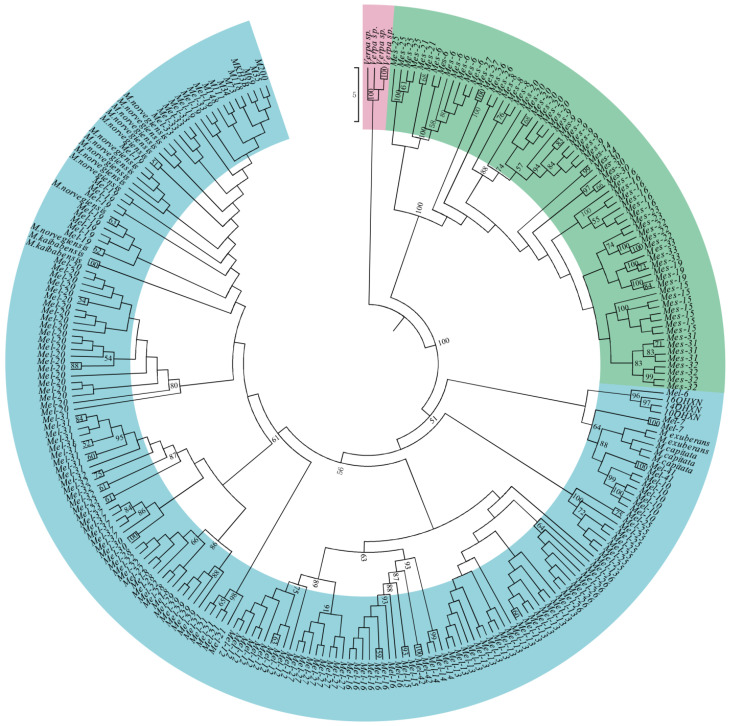
Maximum likelihood phylogenetic tree of *Morchella* based on the ITS, *EF1-α*, *RPB1* and *RPB2* data set. Bootstrap support values based on 1000 pseudoreplicates of the data (>50%) are indicated above each internode. Pink stands for the outgroup, green represents the Esculenta Clade, and blue is the Elata Clade.

**Figure 6 genes-13-01806-f006:**
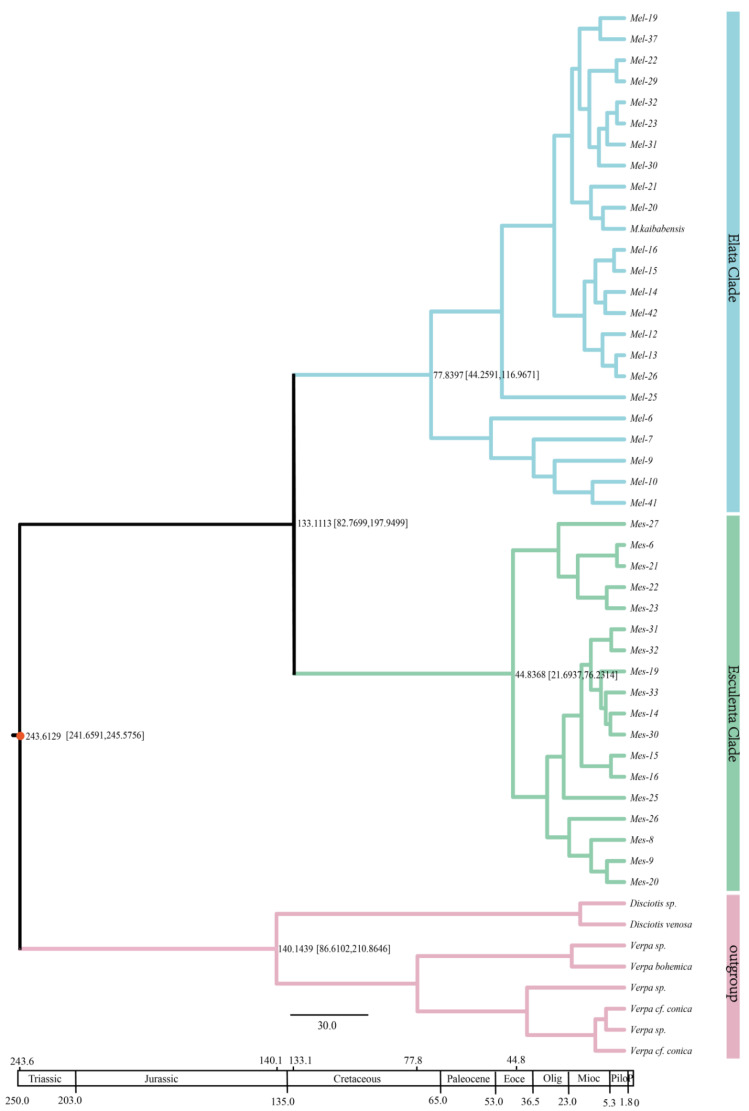
The MCC chronogram is topologically concordant with phylogenies of the Esculenta and Elata Clades that were inferred from the combined three-locus data sets (*EF1-α*, *RPB1* and *RPB2*). The differentiation time of Morchellaceae was used as the calibration point (O’Donnell, 2011). The color pink stands for the outgroup, *Verpa* sp. And *Disciotis* sp.; green represents the Esculenta Clade; and the Elata Clade is shown in blue. The bars of 95 HPD for each divergence are marked on the branch.

**Table 1 genes-13-01806-t001:** Barcode primers and reaction conditions.

DNA Region	Primer Pairs	Primer Sequences (5′-3′)	Thermal Cycling Conditions	References
*EF1*-*α*	*EF1*-*α*-F	ACTCCTAAGTACTATGTCACCGTCATT	94 °C 2 min, 35 cycles (94 °C 1 min, 55 °C 40 s, 72 °C 1 min), 72 °C 10 min	[24]
	*EF1*-*α*-R	TGGAGAGGAAGACGGAGAGGCTT		
*RPB1*	*RPB1*-F	TATATCACGTCGGTATGTATCCACTC	94 °C 2 min, 35 cycles (94 °C 1 min, 55 °C 40 s, 72 °C 1 min), 72 °C 10 min	[25]
	*RPB1*-R	ATTTGCTCGGATGATCTCAG		
*RPB2*	*RPB2*-F	TAGGTAGGTCCCAAGAACACC	94 °C 2 min, 35 cycles (94 °C 1 min, 55 °C 40 s, 72 °C 1 min), 72 °C 10 min	[25]
	*RPB2*-R	GATACCATGGCGAACATTCTG		
ITS	ITS-1	TCCGTAGGTGAACCTGCGG	94 °C 2 min, 35 cycles (94 °C 1 min, 55 °C 40 s, 72 °C 1 min), 72 °C 10 min	[26]
	ITS-4	TCCTCCGCTTATTGATATGC		

**Table 2 genes-13-01806-t002:** The variability characteristics of single DNA molecular barcoding genes in *Morchella*.

Gene	Length (bp)	No. SNPs	% SNP	No. inDels	% No. Variable Sites	Intraspecific Distance (Mean)	Interspecific Distance (Mean)	NJ Rate (%) P-Distance	NJ Rate (%) K2P	PWG-Distance Rate (%) K2P
*RPB1*	420	94	22.38	2	22.85	0.00056	0.0673	33.33	35.56	53.33
*RPB2*	391	86	21.99	0	21.99	0.00057	0.0026	37.78	40.00	63.04
*ITS*	452	125	27.65	78	44.91	0.00062	0.1347	48.89	44.44	57.78
*EF1-α*	312	76	24.36	18	30.13	0.00040	0.077	46.67	42.22	53.33

Length, each single region’s aligned sequence length; No. SNPs, the number of SNPs; % SNP, percentage SNP calculated as the number of SNPs in relation to the longest sequence length; No. inDels, the number of insertions/deletions; % No. variable sites, the ratio of the sum of SNP and insertions/deletions sites to the longest sequence length; interspecific distance (mean), the barcoding gap between species; intraspecific distance (mean), the barcoding gap within species; rate (%), percentage of successful discrimination species calculated as the number of successful discrimination species in relation to the total species; NJ, tree-building method (neighbor-joining tree); PWG-distance, intraspecific and interspecific genetic distance approach; P-distance and K2P represent different model parameters.

## Data Availability

The supported data was from the Appendix A.

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
