# Peer review of "DNA Barcoding and Species Classification of *Morchella"

_genes, 2022, doi:10.3390/genes13101806_

Round 1

Reviewer 1 Report

The manuscript show important, abrangent and necessary study of barcode for Morchella. 

The text is clear (some typos bellow), the analysis correct (except by Parsimony - see bellow) and the findings is important for species identification for the group.

Just 12 of 270 are new sequences and it is not clear if the new sequences impact the results of just the use of genbank acessions.

Some typos found in the text:

Line 66: In present, some instead In present, Some...

Line 83: mono-phyly  instead mon-ophyly...

Line 88: molecular instead molecualr...

Table 1. Insert the reference for the primers

Revise the scientific names in italics in all text.

Figure 6 needed the bars of 95 HPD for each divergence

General aspects:

It is unclear why to conduct Phylogenetic analysis with RAxML and Parsimony if the second Barcode Method is a Phylogenetic construction by NJ. Mega is not appropriated to run Parsimony. I suggest using PAUP (which could be done in the CIPRES portal) or in TNT, or removing all indication of Parsimony analyses since there is no discussion of that in the text.

The model test was used only in the concatenated dataset? The models of the individual markers was not measured?

In the Diversification time estimates, I suggest testing Yule and Birth and Death models and evaluating which is more appropriated for the data. It is not clear why estimate the diversification without any biogeographical optimization to discuss the results.

Reviewer 2 Report

The manuscript describes the identification of Morchella spp. using DNA barcoding.

It is a good work considering the number of samples used and combination of molecular markers (EF1-α, RPB1, RPB2 and ITS) deployed to address the problem. 

English language can be improved in methodology and in result section. 

Author Response

Dear reviewers:
    We sincerely thank you for your comments and suggestions. Our comments on your changes are highlighted in MS word. Thanks again.